# On Difference Pattern Synthesis for Spherical Sensor Arrays [note 1]

**DOI:** 10.3390/s24072361

**Published:** 2024-04-08

**Authors:** Zhijiang Huang, Maolin Chen, Xianglu Li, Shunqin Xie, Guoning Ma, Jie Tian

**Affiliations:** Institute of Electronic Engineering, China Academy of Engineering Physics, Mianyang 621999, China; zhijianghuang_iee@caep.cn (Z.H.); cmo@caep.cn (M.C.); bluelxl@163.com (X.L.); suntrain@caep.com (S.X.); maguoning99@caep.cn (G.M.)

**Keywords:** spherical sensor array, difference pattern, spherical harmonics, phase-mode processing, sidelobe suppression

## Abstract

An innovative method for synthesizing optimum difference patterns of the spherical sensor array is introduced, along with a sidelobe tapering technique. Firstly, we suggest employing the spherical harmonics of degree ±1 to synthesize the spherical array difference pattern; secondly, we study the mapping relationship between the difference pattern of the spherical sensor array and the difference pattern of the uniformly spaced linear array (ULA) with odd-numbered elements; finally, we enhance the Zolotarev difference pattern, which is a counterpart to the Dolph–Chebyshev sum pattern that traditionally allows synthesis only for ULA with even-numbered elements. Our modification extends its applicability to synthesize difference patterns for ULA with odd-numbered elements. Leveraging the optimal difference pattern, a generalized Bayliss difference pattern synthesis method designed for the ULA with odd-numbered elements is further proposed. To illustrate the effectiveness of our approach, we present several design examples through experimental simulation.

## 1. Introduction

Spherical sensor arrays have been extensively investigated within the phased array antenna and the acoustic array community for several decades, spanning a diverse range of applications. These applications cover super-resolution direction finding [1,2,3], source localization [4,5], mobile communications, satellite communications [6,7,8], radar [9,10], and numerous others [11]. When dealing with targets distributed in a broad area of three-dimensional space, a spherical sensor array emerges as the optimal choice due to its superior performance and the deployment efficiency of sensors. This preference is attributed to its omnidirectional beam-steering capability across the entire three-dimensional space [12,13].

This paper focuses on the mono-pulse technique applied to spherical sensor arrays. The mono-pulse technique is used for direction of arrival (DOA) estimation of a target. In Figure 1, the sum and difference patterns are symmetrical and anti-symmetrical about the z-axis. The ratio of the received signal to the sum and difference patterns varies with the angle between the target and the *z*-axis, which can be used for DOA estimation and tracking. The accuracy of DOA estimation is related to the width of the main lobe, and the narrower the main lobe, the higher the estimation accuracy. In order to reduce the impact of interference signals, both sum and difference patterns also need to meet low sidelobe constraints. The main lobe width mainly depends on the array size, the optimization range of this metric does not vary much for a defined array. Therefore, low sidelobes are often the goal pursued by various methods for sum- and difference-pattern synthesis, and it is also the main focus of this paper.

Numerous direct sum- and difference-pattern synthesis methods exist for uniformly spaced linear arrays (ULAs) and uniformly spaced planar arrays, including the Dolph–Chebyshev sum pattern and the Zolotarev difference pattern [14]. The Zolotarev difference pattern, developed by McNamara using Zolotarev polynomials, represents the optimal difference pattern for ULAs with even-numbered elements [15,16,17]. In this context, the optimal difference pattern denotes the pattern with the narrowest first null width and the largest normalized difference slope on boresight for a specified sidelobe level (SLL). However, McNamara’s method is limited to synthesizing arrays with even-numbered elements [18]. In an effort to overcome this limitation, S.R. Zinka proposed a generalized Bayliss difference-pattern synthesis method by altering the array factor zeros of the Zolotarev difference pattern; nevertheless, this method also confines syntheses to arrays with even-numbered elements [19].

Unlike linear and planar arrays, synthesizing patterns for spherical arrays and general conformal arrays pose significant challenges and may even be deemed impossible. This difficulty arises from the non-uniform spacing of elements and varying element orientations typical in conformal arrays. Consequently, numerical-pattern synthesis methods for conformal arrays have been a focal point, with numerous efficient approaches proposed over the last two decades. Various techniques, such as Genetic algorithms [20], particle swarm optimization [21], convex optimization [22,23], and adaptive array theory [24] have been explored for conformal-array sum-pattern synthesis. Simultaneously, iterative least-squares [25], convex optimization [9,26], and modified differential evolution algorithms [27] have been examined for conformal-array difference-pattern synthesis. In general, numerical synthesis methods do not guarantee optimal results and often involve substantial computational complexity, given that the optimization problem is typically solved iteratively.

The phase-mode processing of circular arrays can be extended to spherical arrays, enabling the expression of the circular/spherical array pattern as the sum of a series of harmonics. These characteristics have facilitated numerous applications for the circular/spherical array, including broadband-pattern synthesis, direction finding, and super-resolution direction finding [28]. Recently, Koretz and Rafaely proposed a Dolph–Chebyshev sum-pattern synthesis method for a spherical sensor array in the phase-mode domain [29]. In the case of a symmetric response, the research was expanded to establish a linear transform between the ULA and the spherical array. This allows the application of specific processing techniques designed for the ULA to be available for spherical arrays [30]. The pattern synthesis method presented in [29,30] can be analyzed into two steps: (1) selecting a specific series of spherical harmonics (degree 0) to synthesize the desired pattern and further reformulating the pattern as the summation of associated Legendre polynomials, and (2) examining the relationship between the associated Legendre polynomials and the polynomial that defines the Dolph–Chebyshev pattern or the generalized sum pattern of the ULA.

In this paper, we present a direct optimal-difference-pattern synthesis method for spherical arrays. Our fundamental approach shares similarities with the sum-pattern synthesis procedure outlined in [29,30]. Initially, we suggest utilizing the spherical harmonics of degree ±1 to synthesize the difference pattern of the spherical array. Subsequently, we establish the mapping relationship between the difference pattern of spherical arrays and the difference pattern of ULAs with odd-numbered elements. Lastly, the Zolotarev difference pattern is introduced and a generalized Bayliss difference pattern synthesis method tailored for ULAs with odd-numbered elements is proposed. This work extends [31] by providing difference patterns of arbitrary sidelobe level and envelope taper.

The structure of this paper unfolds as follows. In Section 2, we review the phase-mode processing and the pattern-synthesis method in the phase-mode domain for the spherical array. Section 3 introduces the proposed difference-pattern synthesis method, while Section 4 provides the simulation results for further illustration. Finally, Section 5 encompasses concluding remarks.

## 2. Background

Consider the weighting function ω(kR,Ω) over the surface of a sphere with the radius R, where k is the wave number and Ω≡(θ,ϕ) denotes the spatial coordinates in a spherical coordinate system. The radiation pattern of the spherical array can be expressed in both the spatial domain and in the phase-mode domain as follows:(1)$$B\left(\Omega \right)=\int_{\Omega^{\prime }\in S^{2}}\omega^{*}\left(kR,\Omega^{\prime }\right)p\left(kR,\Omega^{\prime },\Omega \right)d\Omega^{\prime }=\sum_{n=0}^{\infty }\sum_{m=-n}^{n}\omega_{nm}^{*}\left(kR\right)p_{nm}\left(kR\right)$$
where $p\left(kR,\Omega^{\prime },\Omega \right)$ represents the response of the sensor located at $\Omega^{\prime }$ to the wavefield impinge from Ω; $\omega_{nm}$ and $p_{nm}$ denote the spherical Fourier transforms of ω(kR,Ω) and $p\left(kR,\Omega^{\prime },\Omega \right)$, respectively. 

In a case of a unit amplitude plane-wave case $p_{nm}$ can be expressed:(2)$$p_{nm}\left(kR\right)=b_{n}\left(kR\right)Y_{nm}^{*}\left(\Omega \right)$$
where $Y_{nm}\left(\Omega \right)=\sqrt{\frac{2n+1}{4\pi }\frac{\left(n-m\right)!}{\left(n+m\right)!}}P_{nm}\left(\cos\theta \right)e^{jm\phi }$ denotes the spherical harmonics of order n and degree m; $P_{nm}\left(\cos\theta \right)$ is the associated Legendre functions; $b_{n}\left(\cdot \right)$ is the mode amplitude of order n and is a function of kR and the sphere configurations. For the cases of the omnidirectional sensor (the spherical array composed of omnidirectional sensors is also referred to as the open-sphere in the acoustics community) and the cardioid sensor, $b_{n}$ can be expressed as presented in [11]:(3)$$b_{n}\left(kR\right)=\left\{ \begin{array}{l} 4\pi i^{n}j_{n}\left(kR\right)\;\;\;\;\;\;\;\;\;\;\;\;\;\;Omnidirectional\;sensor\\ 4\pi i^{n}\left(j_{n}\left(kR\right)-ij_{n}^{\prime }\left(kR\right)\right)\;\;\;\;\;\;\;\;\;Cardioid\;sensor \end{array}\right.$$
where $j_{n}\left(\cdot \right)$ and $j_{n}^{\prime }\left(\cdot \right)$ represent the spherical Bessel function of first kind and its derivation, respectively.

By substituting Equation (2) into Equation (1), and assuming that array is of finite order N such that $\omega_{nm}=0$ for n>N, the radiation pattern in the phase-mode domain can be expressed:(4)$$\begin{array}{l} B\left(\Omega \right)=\sum_{n=0}^{N}\sum_{m=-n}^{n}\omega_{nm}^{*}\left(kR\right)b_{n}\left(kR\right)Y_{nm}^{*}\left(\Omega \right)\\ \;\;\;\;\;\;\;\;=\sum_{n=0}^{N}\sum_{m=-n}^{n}\omega_{nm}^{*}\left(kR\right)b_{n}\left(kR\right)\sqrt{\frac{2n+1}{4\pi }\frac{\left(n-m\right)!}{\left(n+m\right)!}}P_{nm}\left(\cos\theta \right)e^{-jm\phi } \end{array}$$

Remark: Equations (2) and (4) are valid for scalar sensors (such as microphone sensors or sonar sensors) whose patterns exhibit rotational symmetry along the radial axis of the sensor. When accounting for the mutual coupling effect, rotational symmetry properties can be approximately satisfied if the elements are distributed on the sphere’s surface according to the spherical t-design or the Coulomb design, refer to [32]. In this paper, the spherical t-design is specifically adopted. The rotational symmetry properties may not be satisfied in the practical application, such as in Ref. [2] and our previous work (Ref. [26]), non-ideal factors in engineering can be obtained using electromagnetic simulation methods and the desired pattern and can be synthesized using numerical iterative optimization algorithms; the methodology proposed in the paper can be applied to pre-theoretical designs and performance evaluation in each case.

For the modal sum pattern with the look direction along the *z*-axis, indicating rotational symmetry around the *z*-axis, only the degree m=0 is considered, and $\omega_{n,0}$ is optimized to obtain the optimal sum pattern, as detailed in [29].

## 3. Proposed Difference Pattern Synthesis Method

### 3.1. Spherical Sensor Array Difference Pattern

The general difference pattern for the look direction along the *z*-axis can be expressed:(5)$$\begin{array}{l} D_{x}\left(\theta ,\phi \right)=\cos\phi D_{\theta }\left(\theta \right)\;\;\;\;\;\;\;\;(a)\\ D_{y}\left(\theta ,\phi \right)=\sin\phi D_{\theta }\left(\theta \right)\;\;\;\;\;\;\;\;(b) \end{array}$$
where $D_{\theta }\left(0\right)=0$, so $D_{x}\left(\theta ,\phi \right)$ is zero in the yz-plane, the function is antisymmetrical about the yz-plane, and the maximum slope of the function is in the xz-plane. The function $D_{y}\left(\theta ,\phi \right)$ exhibits similar properties.

In order to construct the modal difference patterns as given in Equation (5) from expression (4), only the degree m=±1 is taken into consideration. In other words, the phase-mode domain weights are $\omega_{nm}=0$ for m≠±1. We denote the modal difference pattern as D(Ω):(6)$$D\left(\Omega \right)=\sum_{n=0}^{N}b_{n}\left(kR\right)\left[\omega_{n,-1}^{*}\left(kR\right)Y_{n,-1}^{*}\left(\Omega \right)+\omega_{n,1}^{*}\left(kR\right)Y_{n,1}^{*}\left(\Omega \right)\right]$$

Given that $Y_{n,-m}\left(\Omega \right)={\left(-1\right)}^{m}Y_{n,m}^{*}\left(\Omega \right)$, if we set $\omega_{n,-1}=-\omega_{n,1}$, then the modal difference pattern (6) simplifies to
(7)$$\begin{array}{l} D\left(\Omega \right)=\sum_{n=1}^{N}b_{n}\left(kR\right)\left[-\omega_{n,-1}^{*}\left(kR\right)Y_{n,1}\left(\Omega \right)+\omega_{n,1}^{*}\left(kR\right)Y_{n,1}^{*}\left(\Omega \right)\right]\\ \;\;\;\;\;\;\;\;\;=\sum_{n=1}^{N}\omega_{n,1}^{*}\left(kR\right)b_{n}\left(kR\right)\left[Y_{n,1}\left(\Omega \right)+Y_{n,1}^{*}\left(\Omega \right)\right]\\ \;\;\;\;\;\;\;\;\;=\cos\phi \sum_{n=1}^{N}2\omega_{n,1}^{*}\left(kR\right)b_{n}\left(kR\right)\sqrt{\frac{2n+1}{4\pi }\frac{\left(n-1\right)!}{\left(n+1\right)!}}P_{n,1}\left(\cos\theta \right)\\ \;\;\;\;\;\;\;\;\;=\cos\phi D_{\theta }\left(\theta \right) \end{array}$$
where $D_{\theta }\left(\theta \right)$ represents the weighted summation of the associated Legendre functions in the expression. Since $P_{nm}\left(\pm 1\right)=0$ for m≥1, we can deduce that $P_{n,1}\left(\cos\theta \right){|}_{\theta =0,\pi }=0$ and $D_{\theta }\left(0\right)$ = $D_{\theta }\left(\pi \right)$ = 0. The function $D_{\theta }\left(\theta \right)$ is weighted with cosϕ so the maximum slope is in the xz-plane and the function is zero in the yz-plane. The expression D(Ω) in Equation (7) provides a difference pattern for the xz-plane, and by setting $\omega_{n,-1}=\omega_{n,1}$, we can obtain a difference pattern for the yz-plane. We now shift our focus to the design of $D_{\theta }\left(\theta \right)$ to achieve the optimal difference pattern for the spherical array.

To further streamline the problem, we reformulate $D_{\theta }\left(\theta \right)$ in a more compact form:(8)$$D_{\theta }\left(\theta \right)=\sum_{n=1}^{N}{\tilde{\omega }}_{n}P_{n,1}\left(\cos\theta \right)={{\tilde{\omega }}_{N}}^{T}p_{N}\left(\theta \right)$$
where ${\tilde{\omega }}_{n}=2\omega_{n,1}^{*}\left(kR\right)b_{n}\left(kR\right)\sqrt{\frac{2n+1}{4\pi }\frac{\left(n-1\right)!}{\left(n+1\right)!}}$, n=1,⋯,N. The column vectors ${\tilde{\omega }}_{N}$ and $p_{N}\left(\theta \right)$ are composed of the elements ${\tilde{\omega }}_{n}$ and $P_{n,1}\left(\cos\theta \right)$, respectively. 

The associated Legendre functions Pnm(cosθ) can be decomposed as in [33]:(9)Pnm(cosθ)=cnmTdn(θ)
where cnm∈ℂ2n×1 is the coefficient vector, and dn(θ)∈ℂ2n×1 is the Fourier series. The coefficient vector cnm can be derived from cn,m−1 and cn,m−2 using the recurrence expressions outlined in reference [33]. For the special case of n=1, cn,1 is an imaginary vector and exhibits conjugate symmetry characteristics, i.e., cnm can be expressed as cn,1=[Jc^n,1∗;0;c^n,1], and J is the exchange matrix. Thus, Pn,1(cosθ) can be decomposed as Pn,1(cosθ)=2ic^n,1Tsn(θ), sn(θ)=[sinθ,sin2θ,⋯,sinnθ]T, and the vector pN(θ) in (8) can be further decomposed as the following:(10)pN(θ)=[P1,1(cosθ),P2,1(cosθ),⋯,PN,1(cosθ)]T=[2ic^1,1T,0T2ic^2,1T,0T⋮2ic^N,1T]sN(θ)=CNsN(θ)
where CN is a real lower-triangular full-rank matrix. By substituting Equation (10) into (8), Dθ(θ) can be expressed:(11)Dθ(θ)=ω˜NTpN(θ)=ω˜NTCNsN(θ)

We will demonstrate that the spherical array difference pattern in (11) is analogous to the difference pattern of the ULA with 2N+1 elements. Considering the ULA with 2N+1 elements (only the standard linear array is considered in this paper, i.e., the elements are spaced uniformly with d=λ/2), the weighting function $a_{n}$ for the difference pattern is antisymmetrical and the excitation of the element at the origin is $a_{0}=0$, so the excited element number is actually 2N. The difference pattern of the ULA with 2N+1 and 2N+2 elements can be expressed:(12)$$\left\{ \begin{array}{l} F_{d}^{2N+1}\left(\psi \right)=\sum_{n=-N}^{N}a_{n}e^{i2\pi dn\sin\theta /\lambda }=\sum_{n=1}^{N}a_{n}\sin\left(n\psi \right)={\left(a^{N}\right)}^{T}s_{N}\left(\psi \right)\;\;\;\;\;\;\;\;\;\;\;\;\;\;\;\;\;\;\;\left(a\right)\\ F_{d}^{2N+2}\left(\psi \right)=\sum_{n=-N}^{N+1}b_{n}e^{i2\pi d\left(n-1/2\right)\sin\theta /\lambda }=\sum_{n=1}^{N+1}b_{n}\sin\left(\left(n-\frac{1}{2}\right)\psi \right)={\left(b^{N+1}\right)}^{T}{\hat{s}}_{N+1}\left(\psi \right)\;\;\;\left(b\right) \end{array}\right.$$
where ${\hat{s}}_{N+1}\left(\psi \right)={\left[\sin\frac{\psi }{2},\sin\frac{3\psi }{2},\cdots ,\sin\left(\frac{L+1}{2}\right)\psi \right]}^{T}$, and $\psi =\frac{2\pi d\sin\theta }{\lambda }$. $b_{n}$, n=−N,⋯,N+1 is the weighting function with a 2N+2-elements ULA difference pattern, which is also antisymmetrical. 

If we set ${C_{N}}^{T}{\tilde{\omega }}_{N}=a^{N}$, then the spherical array difference pattern in Equation (11) and the ULA difference pattern in Equation (12) (a) become identical with the transformation ψ=2πdsinθ/λ. Since $C_{N}$ is a full-rank matrix, ${\tilde{\omega }}_{N}$ can be uniquely solved. Subsequently, $\omega_{nm}\left(kR\right)$ and the weighting function ω(kr,Ω) will be determined sequentially.

When the look direction steers away from the *z*-axis, the desired pattern can be obtained using the following expression:(13)$$D\left(\Omega \right)=\sum_{n=0}^{N}\sum_{m=-n}^{n}\omega_{nm}^{\mathrm{rot}*}\left(kR\right)b_{n}\left(kR\right)Y_{nm}^{*}\left(\Omega \right)$$
where $\omega_{nm}^{\mathrm{rot}}$ is given by
(14)$$\begin{array}{l} \omega_{nm}^{rot}=\left[\begin{array}{c} \omega_{n,-n}^{rot}\\ \omega_{n,-n+1}^{rot}\\ \vdots \\ \omega_{n,n}^{rot} \end{array}\right]=\left[\begin{array}{cccc} D_{-n,-n}^{n} & D_{-n,-n+1}^{n} & \cdots & D_{-n,n}^{n}\\ D_{-n+1,-n}^{n} & D_{-n+1,-n+1}^{n} & \cdots & D_{-n+1,n}^{n}\\ \vdots & \vdots & \ddots & \vdots \\ D_{n,-n}^{n} & D_{n,-n+1}^{n} & \cdots & D_{n,n}^{n} \end{array}\right]\left[\begin{array}{c} 0\\ \omega_{n,1}\\ 0\\ \omega_{n,1}\\ 0 \end{array}\right]\\ \;\;\;\;\;\;=\omega_{n,-1}{\left[\begin{array}{cccc} D_{-n,1}^{n} & D_{-n+1,1}^{n} & \cdots & D_{n,1}^{n} \end{array}\right]}^{T}+\omega_{n,1}{\left[\begin{array}{cccc} D_{-n,-1}^{n} & D_{-n+1,-1}^{n} & \cdots & D_{n,-1}^{n} \end{array}\right]}^{T} \end{array}$$
where $D_{{mm}^{\prime }}^{n}≜D_{{mm}^{\prime }}^{n}\left(\alpha ,\beta ,\gamma \right)$ denotes the Wigner-D function, and (α,β,γ) represents the Euler rotation angle from the *z*-axis to the look direction [11].

### 3.2. Zolotarev Difference Pattern of 2N+1 Elements

As a counterpart to the Dolph–Chebyshev sum pattern, the Zolotarev difference pattern is an optimal-difference pattern for the even-numbered ULA developed by McNamara using the Zolotarev polynomials [16]. The procedure for synthesizing the Zolotarev difference pattern for a ULA with 2N+2 elements is summarized as follows:


(1)For a specified sidelobe ratio (SLR) or the main-lobe width, the Jacobi modulus parameter m, which is related to the specified SLR or main-lobe width, is calculated. Subsequently, the Zolotarev polynomial $Z_{2N+1}\left(x,m\right)$ is evaluated using the numerical method, and its expansion in the standard polynomial form is obtained:(15)$$Z_{2N+1}\left(x,m\right)=\sum_{n=1}^{N+1}z_{n}x^{2n-1}$$


For the computational aspects of the procedure, detailed information can be found in [16]. Knowledge of the elliptical integrals, the Jacobi module and Jacobi eta, zeta, and elliptical functions is essential. Open-source tools for calculating these functions are accessible [34]. Figure 2 provides an example of Zolotarev polynomials $Z_{11}\left(x\right)$, the function oscillates between −1 and 1 when x∈[−1,1].


(2)Let x=sin(ψ/2)/sin(πd/λ), and substitute it into the above polynomial, let $c_{n}=z_{n}/{\left(\sin\left(\pi d/\lambda \right)\right)}^{2n-1}$, then the desired Zolotarev difference pattern can be expressed:(16)$$Z_{2N+1}\left(\psi \right)=\sum_{n=1}^{N+1}c_{n}\sin^{2n-1}\left(\frac{\psi }{2}\right)$$



(3)Equate $F_{d}^{2N+2}\left(\psi \right)$ in Equation (12) (b) to $Z_{2N+1}\left(\psi \right)$ and determine the coefficient $b_{n}$, then, the element excitation can be calculated from $b_{n}$.


However, the Zolotarev difference pattern is only available for the even-numbered ULA. When comparing the 2N+1 and 2N+2 elements ULA difference patterns given by Equation (12) (a) and (12) (b), respectively, the primary distinction lies in the harmonics series. The difference pattern $F_{d}^{2N+1}\left(\pm \pi \right)=0$ is fixed, whereas $F_{d}^{2N+2}\left(\pm \pi \right)$ is variable. For the Zolotarev difference pattern of the even-numbered ULA, $\left|F_{d}^{2N+2}\left(\psi \right)\right|{|}_{\psi =\pm \pi }$= $\left|Z_{2N+1}\left(\psi \right)\right|{|}_{\psi =\pm \pi }$=1. It should also be noted that the number of roots in the visible region for the optimum difference pattern $F_{d}^{2N+1}\left(\psi \right)$ and $F_{d}^{2N+2}\left(\psi \right)$ is the same. Consequently, the optimal difference pattern $F_{d}^{2N+1}\left(\psi \right)$ can also be derived from $Z_{2N+1}\left(\psi \right)$.

Let x=sin(βψ/2)/sin(πd/λ) (where β is the parameter to be solved), such that $F_{d}^{2N+1}\left(\psi \right){|}_{\psi =-\pi }$ and $F_{d}^{2N+1}\left(\psi \right){|}_{\psi =\pi }$ will be mapped to the minimum root $x^{1}$ and the maximum root $x^{2N+1}$ of $Z_{2N+1}\left(x\right)$, respectively. In other words, $x^{1}=\sin\left(-\beta \pi /2\right)/\sin\left(\pi d/\lambda \right)$ and $x^{2N+1}=$ sin(βπ/2)/sin(πd/λ). Thus, β can be determined by solving $\beta =2\sin^{-1}\left(x^{2N+1}\right)/\pi$.

The optimal difference pattern for the ULA with 2N+1 elements can now be obtained using the same procedure as described above for the ULA with 2N+2 elements, with the only difference being the mapping of ψ to x in step 2. 

### 3.3. The Generalized Bayliss Difference Pattern of 2N+1 Elements

In [19], a generalized Bayliss $\bar{n}$ array distributions method was introduced for even-numbered ULA by altering the array factor zeros of the Zolotarev difference pattern. The updated array factor zeros are defined as the following:(17)$$\psi_{n}^{Bayliss}=\left\{ \begin{array}{l} \sigma^{B}\psi_{n}^{Zolatarev}\;\;\;\;n\leq \bar{n}\\ \psi_{n}^{B}\;\;\;\;\;\;\;\;\;\;\;\;\;\;\;\;\;\;n\geq \bar{n} \end{array}\right.$$
where $\psi_{n}^{Zolatarev}$ and $\psi_{n}^{Bayliss}$ denote the array factor zeros of Zolotarev and the generalized Bayliss difference pattern, respectively. The so-called dilation factor $\sigma^{B}$ and the far-end zeros $\psi_{n}^{B}$ are defined:(18)$$\begin{array}{l} \psi_{n}^{B}=\pm \left(n+\frac{\alpha +1}{2}\right)\frac{2\pi }{M},\;\;n=\bar{n},\bar{n}+1,\cdots ,ceil\left[\left(M-2\right)/2\right]\\ \sigma^{B}=\frac{\psi_{\bar{n}}^{B}}{\psi_{\bar{n}}^{Zolatarev}} \end{array}$$
where M represents the number of elements, α is the parameter controlling the envelop tapering. $\psi_{n}^{Bayliss}$ is obtained from the line source difference pattern, and thus can be utilized for both even-numbered and odd-numbered ULAs. However, this study only considered an even-numbered ULA since the Zolotarev difference pattern for the odd-numbered ULA was not available [19].

Now that we have derived the difference pattern for the odd-numbered ULA, the generalized Bayliss pattern can also be obtained from (16), with the caveat that the end zeros should be replaced with ±π. Given that the spherical array difference pattern (11) and the ULA difference pattern (12) (a) are identical through the transformation ψ=2πdsinθ/λ, the Zolotarev difference pattern and the generalized Bayliss difference pattern for the spherical array can also be synthesized using the same transformation relationship.

## 4. Simulations

In this section, we present some design examples for the ULA, spherical aperture, and spherical sensor array through computer simulations. 

### 4.1. The Difference Pattern of the ULA

Firstly, we consider the case of the ULA. The array is composed of 17 elements (N=8), with element spacing as half-wavelength. In order to verify that the proposed method achieves optimal results, a convex optimization method is used here as a comparison. The problem of synthesizing a difference pattern for the ULA with the narrowest beamwidth and largest normalized difference slope on boresight for a specified sidelobe level can be formulated:(19)$$\begin{array}{l} \underset{w}{\max}\left\{\frac{d}{d\psi }\left[w^{T}s_{N}\left(\psi \right)\right]{|}_{\psi =0}\right\}\\ subject\;to\;\;\left\{{\left|w^{T}s_{N}\left(\psi_{j}\right)\right|}^{2}\right\}\leq S\left(\psi_{j}\right),\;\psi_{j}\in \Theta ,\;j=1,\cdots ,J \end{array}$$
where Θ represents the sidelobe region; $\psi_{j}\in \Theta ,\;\;j=1,\cdots ,J$ are the discrete angular grid points representing Θ; J is the number of inequality constraints; $S\left(\psi_{j}\right)$ is the upper bound for the sidelobe level in the direction $\psi_{j}$. The pattern synthesis formulations in (19) can be reformulated as a convex optimization problem [35] and be efficiently solved using numerical methods with existing free software, such as the CVX Toolbox with version 2.2 [36]. The convex optimization method ensures that the optimal is reached; thus, the pattern given by CVX is optimal.

We set the sidelobe region and N in (19) to be the same as the Zolotarev difference pattern, solved the convex optimization problem, and plotted the pattern in Figure 3. The SLRs are set to −25 dB and −35 dB, respectively. The desired SLRs are achieved using the two methods, and the convex optimization-based difference pattern is overlapped with the Zolotarev difference pattern for the two simulation conditions. The figure confirms that both the proposed method and the numerical method in Ref. [35] achieve the same optimum difference pattern with equal sidelobe.

In Figure 4, the algorithm proposed in this paper synthesizes the Zolotarev pattern for ULA with 17 elements, while the method described in reference [16] synthesizes the Zolotarev pattern for ULA with 18 elements. The sidelobe constraint for both cases is set to 25 dB, and the polynomial utilized is the Zolotarev polynomial $Z_{17}\left(x\right)$. It is evident that the ULA with 18 exhibits a narrower first null beamwidth and a larger normalized difference slope than the ULA with 17 elements, owing to its larger array aperture. The array factor zeros of the two arrays appear alternately.

In the next example, the algorithm proposed in this paper synthesizes the generalized Bayliss pattern for ULA with 17 elements, while the method described in reference [16] synthesizes the generalized Bayliss pattern for ULA with 18 elements. The sidelobe constraint for both cases set to 25 dB. Figure 5 illustrates the generalized Bayliss difference patterns for ULA of 17 elements and 18 elements for different values of α, assuming N=8 and $\bar{n}=4$. As observed, for a given α, the two arrays exhibit similar envelop tapering, and the sidelobe closer to ψ=0 are comparable to the corresponding Zolotarev difference pattern. The tapering rate of the far-end sidelobes is correlated with α, with a higher α resulting in a greater tapering rate.

### 4.2. The Difference Pattern of the Spherical Aperture

Now, let us return to the design of the spherical array difference pattern. The Zolotarev difference pattern for a spherical array with N=8 is depicted in Figure 6, assuming SLR=25 dB, the pattern resembles the ULA with 17 elements shown in Figure 4, albeit with different horizontal axes.

The generalized Bayliss difference patterns for the spherical array with α=0.5 and α=1 are depicted in Figure 7 and Figure 8, respectively. It is evident that the patterns resemble those of the ULA with 17 elements shown in Figure 5a,b, although the horizontal axes differ.

In the next example, the Zolotarev difference patterns with the look direction towards (θ,ϕ)=(30°,45°) are presented in Figure 9. These patterns are obtained through coordinate rotation in the phase-mode domain, as given by Equation (13), and the original difference patterns are shown in Figure 8.

The excitation amplitude functions corresponding to the omnidirectional sensor array and the cardioid sensor array sphere configurations are depicted in Figure 10, Figure 11 and Figure 12, assuming kR=7. It can be observed that the excitation amplitude function of the omnidirectional sensor array is symmetrical about the equator for both the Zolotarev difference pattern and the generalized Bayliss difference pattern. In the case of the cardioid sensor array, the main contributions of the excitation amplitude function appear in the upper hemisphere, and the power is concentrated for the Bayliss difference pattern, while the power is not concentrated for the Zolotarev difference pattern.

### 4.3. The Difference Pattern of the Spherical Sensor Array

Finally, a comparison is made between beamforming in the spherical harmonics domain and the spatial domain. A spherical array is considered, consisting of 144 nearly uniform sampling elements distributed on the sphere surface following the spherical-t design [37], as illustrated in Figure 13. The weights of the elements are computed using the method described in [29], where the optimal array weights are sampled to calculate the weights’ values at the sensor positions for the two types of sensors. 

The space-domain Zolotarev difference pattern and the generalized difference pattern with α=0.5 and α=1 for spherical arrays composed of omnidirectional sensors and cardioid sensors are illustrated in Figure 14. The patterns in Figure 6b, Figure 7b, and Figure 8b are utilized as reference patterns. It can be observed that the space-domain difference pattern for the spherical array composed of omnidirectional sensors and cardioid sensors closely resembles the reference patterns, which are synthesized in the phase-mode domain.

## 5. Conclusions

Building upon the phase-mode processing of spherical array and the application of Zolotarev polynomials, this paper introduces a method for synthesizing Zolotarev and generalized Bayliss difference patterns for the spherical sensor arrays and odd-numbered ULAs. Several design examples are provided, utilizing the experimental simulations to validate the efficacy of the proposed method. The main contribution of this paper lies in the theoretical modeling and performance analysis applied to the early stage of array design, and the synthesis results can be used as the initial value of the numerical synthesis methods when dealing with various non-ideal factors. We focus on the common case of scalar sensor arrays such as microphone sensors or sonar sensors. When the vector spherical harmonics are adopted, the results can be extended to vector sensor arrays such as electromagnetic sensors.

## Figures and Tables

**Figure 1 sensors-24-02361-f001:**
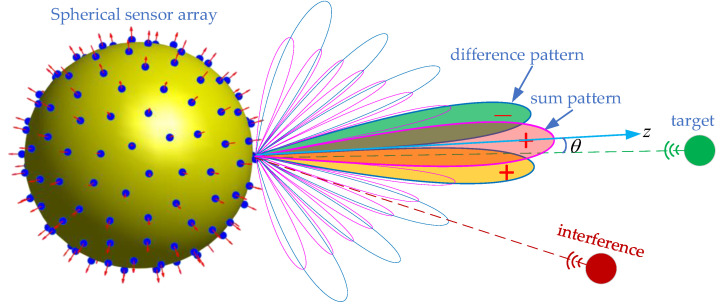
Principle of DOA estimation based on sum- and difference-patterns.

**Figure 2 sensors-24-02361-f002:**
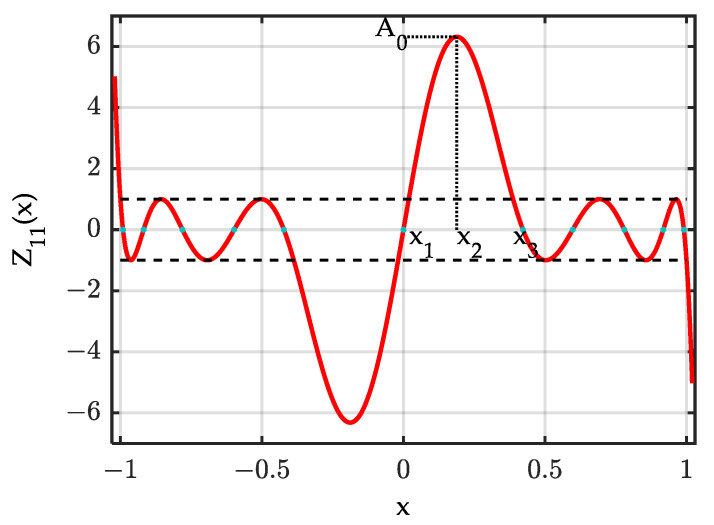
Representation of Zolotarev polynomial $Z_{11}\left(x\right)$.

**Figure 3 sensors-24-02361-f003:**
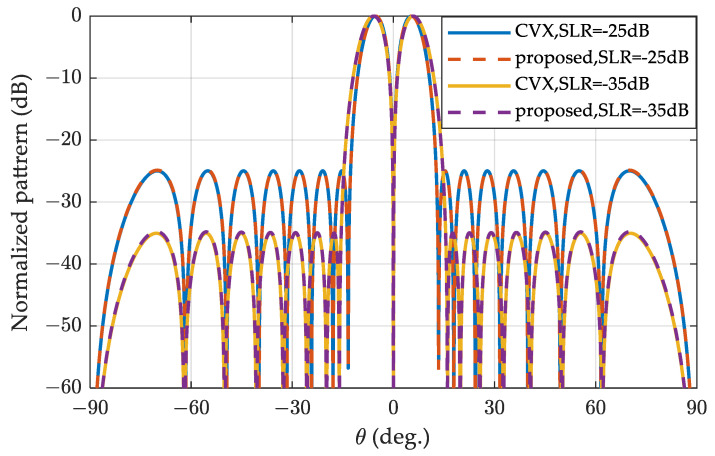
Zolotarev and convex optimization difference patterns for SLR = 25 dB and SLR = 35 dB, respectively.

**Figure 4 sensors-24-02361-f004:**
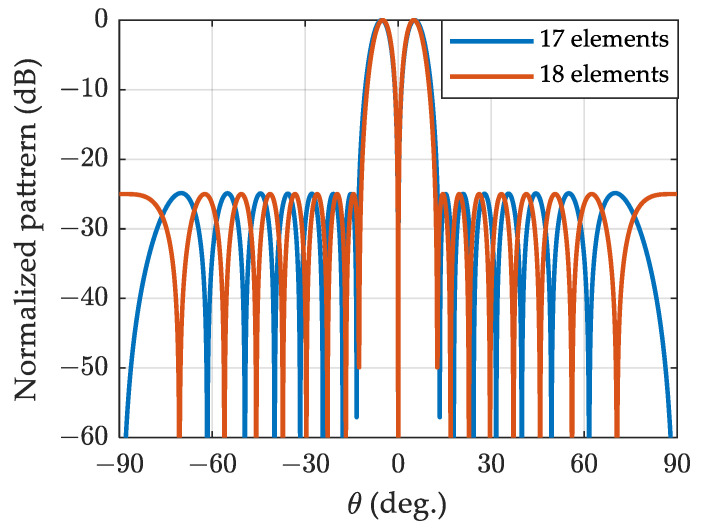
Zolotarev difference patterns of SLR=25 dB for ULA 17 and 18 elements.

**Figure 5 sensors-24-02361-f005:**
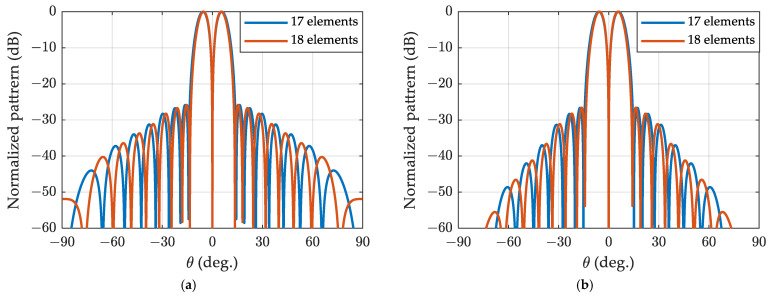
The generalized Bayliss difference patterns with $\bar{n}=4$ for ULA of 17 and 18 elements: (**a**) α=0.5; (**b**) α=1.

**Figure 6 sensors-24-02361-f006:**
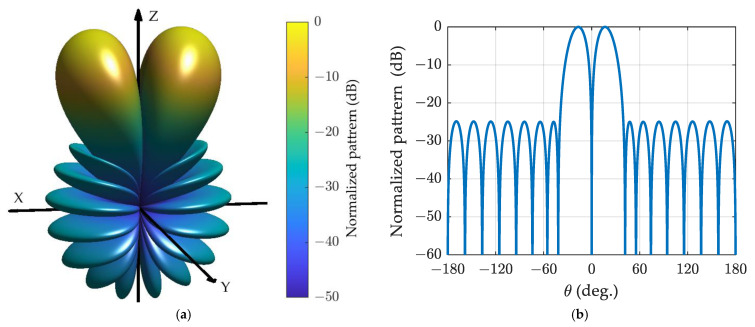
The Zolotrarev difference pattern for spherical array with *N* = 8 and SLR = 25 dB: (**a**) three-dimensional plot; (**b**) cut plot.

**Figure 7 sensors-24-02361-f007:**
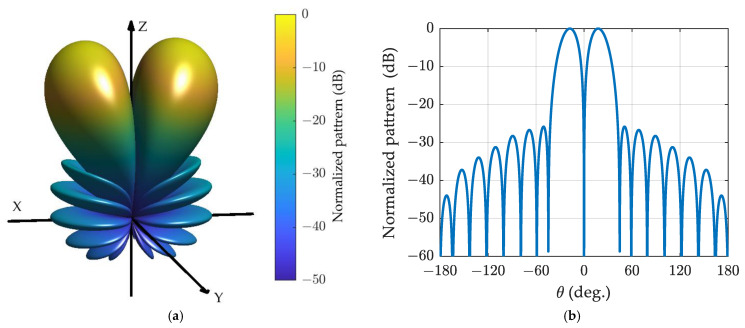
The generalized Bayliss difference pattern for spherical array with *N* = 8, $\bar{n}=4$, α=0.5, and SLR = 25 dB: (**a**) three-dimensional plot; (**b**) cut plot.

**Figure 8 sensors-24-02361-f008:**
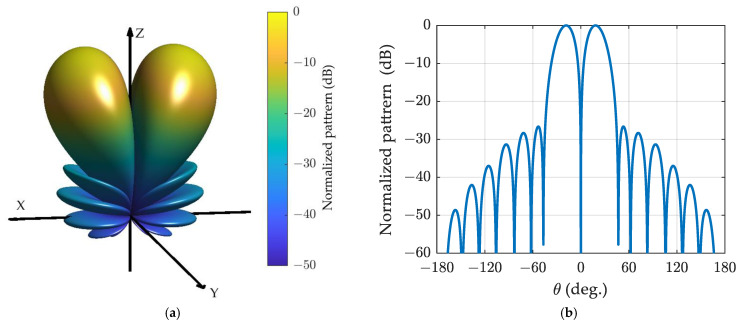
The generalized Bayliss difference pattern for spherical array with *N* = 8 $\bar{n}=4$, α=1, and SLR = 25 dB: (**a**) three-dimensional plot; (**b**) cut plot.

**Figure 9 sensors-24-02361-f009:**
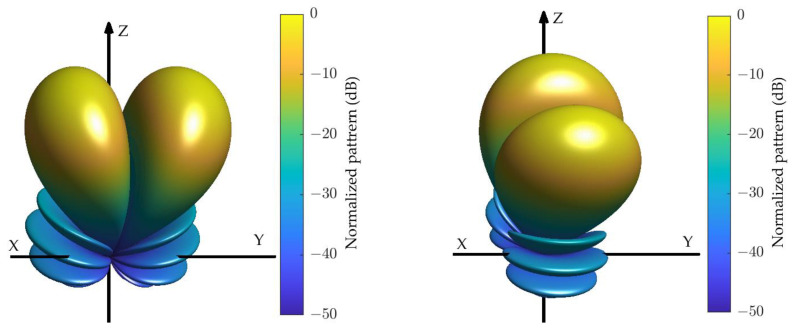
The difference patterns with the look direction towards (θ,ϕ)=(30°,45°).

**Figure 10 sensors-24-02361-f010:**
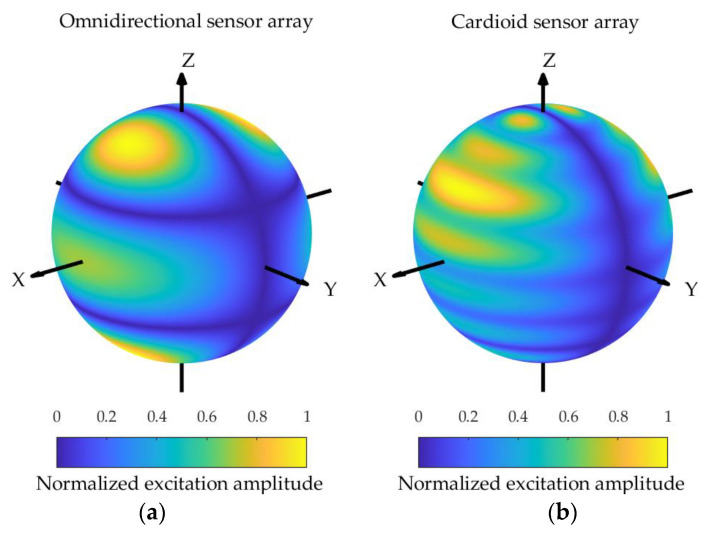
The excitation functions for the spherical array composed of omnidirectional sensors (**a**) and cardioid sensors (**b**) corresponding to the pattern shown in Figure 6.

**Figure 11 sensors-24-02361-f011:**
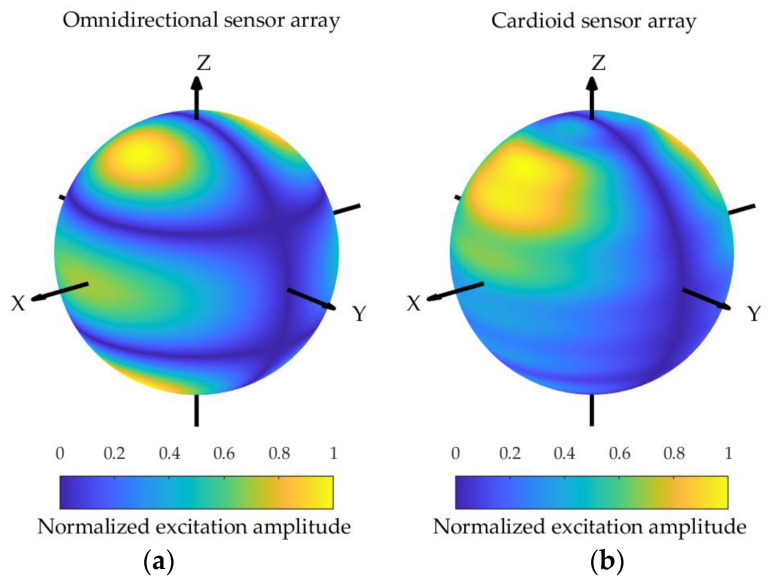
The excitation functions for the spherical array composed of omnidirectional sensors (**a**) and cardioid sensors (**b**) corresponding to the pattern shown in Figure 7.

**Figure 12 sensors-24-02361-f012:**
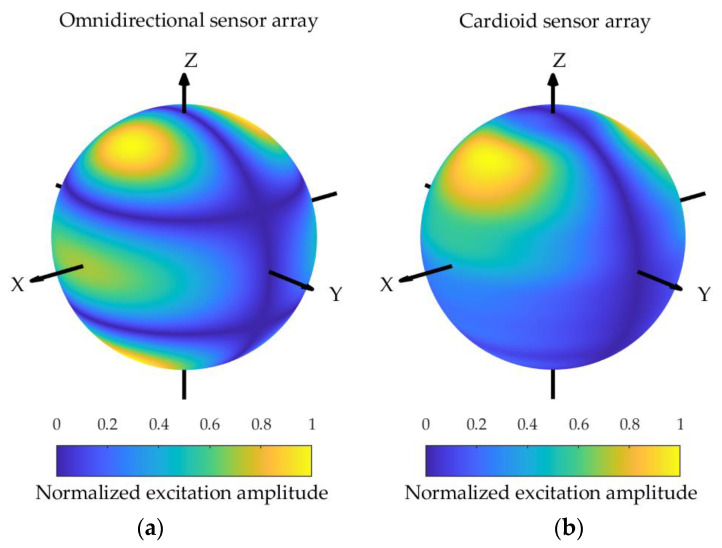
The excitation functions for the spherical array composed of omnidirectional sensors (**a**) and cardioid sensors (**b**) corresponding to the pattern shown in Figure 8.

**Figure 13 sensors-24-02361-f013:**
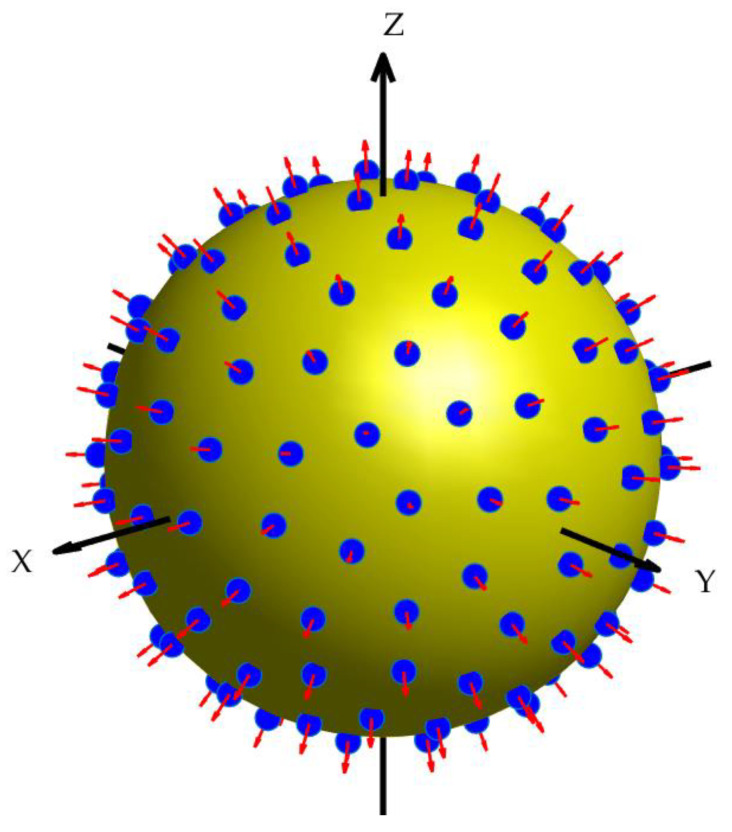
Spherical array with 144 elements.

**Figure 14 sensors-24-02361-f014:**
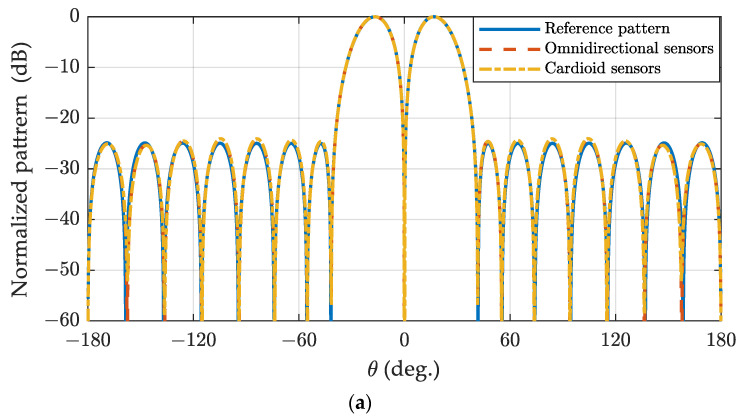
The space-domain Zolotarev difference pattern (**a**) and the generalized difference pattern with α=0.5 (**b**) and α=1 (**c**) for various spherical sensor configurations with 144 nearly uniform sampled elements.

## Data Availability

Data are contained within the article.

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
