# Peer review of "On Difference Pattern Synthesis for Spherical Sensor Arraysâ€"

_sensors, 2024, doi:10.3390/s24072361_

Round 1
Reviewer 1 Report
Comments and Suggestions for Authors
This paper proposes a method for synthesizing optimum difference patterns of the spherical sensor array along with a sidelobe tapering technique. Several design examples through experimental simulation have been presented to illustrate the effectiveness of the proposed method. My main concerns are listed as following:
1. In the Abstract part, the authors directly introduces the proposed method. The research motivation should be introduced firstly.
2. In the lines 36 and 37 of Page 1, “uniformly spaced linear and planar arrays” stands for “(ULAs)”. However, in the line 256, “uniform linear array” stands for the “(ULA)”. The authors should describe these abbreviations clearly.
3. In the line 83 of Page 2, “In section 2, We review …” should be corrected as “In Section 2, we review …”.
4. In the line 89 of Page 2, what does the parameter r stands for in w(kr,\Omega)? What is difference between the parameter r and the parameter R.
5. In the line 94 of Page 2, what does the parameter r stands for in p(kr,\Omega^{\prime},\Omega)?
6. In the equation (12)(a), what does “exp” mean? Please describe it clearly similar to (12)(b).
7. In Figure 3, Does the CVX method produce the optimal values? The patterns of the proposed method are the same as those of the CVX method. What is the reason? Why they have the same patterns?
8. In the simulations, why the proposed method is not be compared with the other current methods? Does the proposed method perform better than other current methods?
9. In the proposed method, why the mutual coupling is not be considered in spherical arrays? For example, the references [2] considered the mutual coupling in spherical arrays.
Comments on the Quality of English LanguageModerate editing of English language required.
Author Response
Dear Reviewer,
We would like to express our gratitude for your thorough review of our paper. Your valuable insights have greatly contributed to the improvement of our manuscript. We have carefully revised the manuscript (ID: sensors-2887480, Title: On Difference Pattern Synthesis of the Spherical Sensor Array) based on your suggestions. We have diligently examined your perceptive comments and made the appropriate corrections, which we hope will meet your approval. The primary revisions in the paper include the following:
Point 1: In the Abstract part, the authors directly introduce the proposed method. The research motivation should be introduced firstly.
Response 1: Thank you for reviewing our manuscript and providing valuable feedback. We appreciate the time and effort you have put into evaluating our work. To address your concerns, we have added Figure 1 in the Abstract part to introduce the research motivation (Please see the Figure 1 and the paragraph above it, “This paper focuses on the mono-pulse technique applied to spherical sensor arrays. The mono-pulse technique is used for direction of arrival (DOA) estimation of a target. In Figure 1, the sum and difference patterns are symmetrical and anti-symmetrical about z axis. The ratio of the received signal to the sum and difference patterns varies with the angle between the target and the z-axis, which can be used for DOA estimation and tracking. The accuracy of DOA estimation is related to the width of the main lobe, and the narrower the main lobe, the higher the estimation accuracy. In order to reduce the impact of interference signals, both sum and difference patterns also need to meet low sidelobe constraints. The main lobe width mainly depends on the array size, the optimization range of this metric does not vary much for a defined array. Therefore, low sidelobes are often the goal pursued by various methods for sum and difference pattern synthesis, it is also the main focus of this paper.”
Point 2: In the lines 36 and 37 of Page 1, “uniformly spaced linear and planar arrays” stands for “(ULAs)”. However, in the line 256, “uniform linear array” stands for the “(ULA)”. The authors should describe these abbreviations clearly.
Response 2: Thank you for reviewing our manuscript and pointing out the issue with the abbreviations description. “ULA” represents “uniformly-spaced linear arrays”, to address your concern, we carefully revised the relative section.
Point 3: In the line 83 of Page 2, “In section 2, We review …” should be corrected as “In Section 2, we review …”.
Response 3: Thank you for providing valuable feedback. We have studied your comments and we have revised the paper carefully and accordingly. “In section 2, We review …” has been corrected as “In Section 2, we review …”
Point 4: In the line 89 of Page 2, what does the parameter r stands for in w(kr,\Omega)? What is difference between the parameter r and the parameter R.
Response 4: Thank you for your insightful feedback on our manuscript. We have studied your comments and we have revised the paper carefully and accordingly. “r” should be modified as “R”, which denotes the radius of the sphere. We have made the necessary modifications, and we have double-checked the equations in the paper to avoid similar misalignment.
Point 5: In the line 94 of Page 2, what does the parameter r stands for in p(kr,\Omega^{\prime},\Omega)?
Response 5: Thank you for your insightful feedback on our manuscript. We have studied your comments and we have revised the paper carefully and accordingly. “r” should be modified as “R”, which denotes the radius of the sphere. We have made the necessary modifications, and we have double-checked the equations in the paper to avoid similar misalignment.
Point 6: In the equation (12)(a), what does “exp” mean? Please describe it clearly similar to (12)(b).
Response 6: Thank you for your valuable feedback on our manuscript. We have studied your comments and we have revised the paper carefully and accordingly. Just like equation (12)(a), “exp” should be modified as “e” in the equation (12)(a). We have made the necessary modifications, and we have double-checked the equations in the paper to avoid similar misalignment.
.Point 7: In Figure 3, Does the CVX method produce the optimal values? The patterns of the proposed method are the same as those of the CVX method. What is the reason? Why they have the same patterns?
Response 7: Thank you very much for your valuable feedback on our manuscript. The CVX method produce the optimal values (the last sentence of the paragraph below equation (19) “The convex optimization method ensures that the optimal is reached, thus the pattern given by CVX is optimal”). The convex optimization method belongs to numerical iterative optimization algorithms, and the proposed method belongs to theoretical calculation methods. Here the CVX method is used as a comparison to show that the proposed method also achieves the optimum. We have modified the paragraph above equation (19), ”In order to verify that the proposed method achieves optimal results, a convex optimization method is used here as a comparison.”.
Point 8: In the simulations, why the proposed method is not be compared with the other current methods? Does the proposed method perform better than other current methods?
Response 8: Thank you very much for your valuable feedback on our manuscript.
In Fig.3 the Zolotarev difference pattern synthesis method for ULAs with odd numbered elements and spherical array proposed in the paper has been compared with the CVX method in ref. [35]. We have made modifications to the paragraph above Figure 3. “The figure confirms that both the proposed method and the numerical method in ref. [35] achieve the same optimum difference pattern with equal sidelobe.”
In Fig.4, the proposed Zolotarev difference pattern for ULA with odd numbered elements has been compared with ULA with even numbered elements. We have made modifications to the paragraph above Figure 4, “In Figure 4, the algorithm proposed in this paper synthesizes the Zolotarev pattern for ULA with 17 elements, while the method described in reference [16] synthesizes the Zolotarev pattern for ULA with 18 elements. The sidelobe constraint for both cases are set to 25 dB, and the polynomial utilized is the Zolotarev polynomial .”
In Fig.5, the proposed generalized Bayliss difference pattern for ULA with odd numbered elements has been compared with ULA with even numbered elements. We have made modifications to the text above Figure 5. “In the next example, the algorithm proposed in this paper synthesizes the generalized Bayliss pattern for ULA with 17 elements, while the method described in reference [16] synthesizes the generalized Bayliss pattern for ULA with 18 elements. The sidelobe constraint for both cases are set to 25dB.”
Point 9: In the proposed method, why the mutual coupling is not be considered in spherical arrays? For example, the references [2] considered the mutual coupling in spherical arrays.
Response 9: Thank you for your valuable feedback on our manuscript. the paragraph below equation (4) gives the remark about the mutual coupling effect (“Equation and are valid for sensors whose patterns exhibit rotational symmetry along the radial axis of the sensor. When accounting for mutual coupling effect, rotational symmetry properties can be approximately satisfied if the elements are distributed on the sphere’s surface according to the spherical t-design or the Coulomb design, refer to [31]. In this paper, the spherical t-design is specifically adopted”). We have extended the remark. (Please see the paragraph below equation (4) “……The rotational symmetry properties may not be satisfied in the practical application, such as ref. [2] and our previous work (ref. [26]), non-ideal factors in engineering can be obtained using electromagnetic simulation methods and the desired pattern and can be synthesized using numerical iterative optimization algorithms, the methodology proposed in the paper can be applied to pre-theoretical designs and performance evaluation in the case.”)
The above corrections we made are high lighted in the revised manuscript.
We hope that these revisions have enhanced the overall quality of our submission and addressed the concerns you raised. We look forward to receiving your further feedback and appreciate your invaluable guidance in helping us refine our work. Thank you once again for your constructive comments and invaluable guidance.
We sincerely hope you will be satisfied with the revised paper.
Best regards,
The Authors
Please see the attachment.

Reviewer 2 Report
Comments and Suggestions for Authors
The authors have introduced a direct approach for optimal difference pattern synthesis; however, the novelty and quality of the study fall short of the journal's criteria.
-
1. The method primarily relies on the amalgamation of two papers (references 29 and 30) that were published over a decade ago. The innovative aspects of the theoretical component remain unclear.
-
2. The manuscript centers predominantly on theory and simulation, lacking sufficient experimental validation for the proposed method. Integrating more empirical evidence would enhance the robustness of the study.
-
3. The motivation behind the research is not explicitly communicated to a general audience. It is recommended to include additional diagrams to elucidate the problem and application scenarios, making the work more accessible and engaging.
The overall quality of the English language in the manuscript is satisfactory.
Author Response
Dear Reviewer,
We would like to express our gratitude for your thorough review of our paper. Your valuable insights have greatly contributed to the improvement of our manuscript. We have carefully revised the manuscript (ID: sensors-2887480, Title: On Difference Pattern Synthesis of the Spherical Sensor Array) based on your suggestions. We have diligently examined your perceptive comments and made the appropriate corrections, which we hope will meet your approval. The primary revisions in the paper include the following:
Point 1: The method primarily relies on the amalgamation of two papers (references 29 and 30) that were published over a decade ago. The innovative aspects of the theoretical component remain unclear.
Response 1: Thank you for reviewing our manuscript and providing valuable feedback.
Traditional array processing mainly focuses on planar arrays, while spherical arrays have emerged as a new research subject in the past two decades, with the theoretical models of spherical arrays continuously evolving. Reference [11] is the first comprehensive monograph we have encountered regarding spherical arrays.
References [29] and [30] investigate the synthesis of sum patterns for spherical arrays, extending the classical Dolph-Chebyshev synthesis method for planar arrays proposed decades ago to the field of spherical arrays. They develop theoretical synthesis methods for sum patterns of spherical arrays, enriching the theoretical modeling methods for spherical arrays.
However, currently, there is no theoretical synthesis method for difference patterns of spherical arrays. Inspired by References [29] and [30], this paper establishes a theoretical synthesis method for difference patterns of spherical arrays, filling this gap in the literature. Additionally, previous classical synthesis methods for difference patterns of linear arrays have only been developed for arrays with even numbered elements. This paper also introduces a synthesis method for difference patterns of linear arrays with odd numbered elements, thus making an innovative breakthrough in theory.
Point 2: The manuscript centers predominantly on theory and simulation, lacking sufficient experimental validation for the proposed method. Integrating more empirical evidence would enhance the robustness of the study.
Response 2: Thank you for reviewing our manuscript and providing valuable feedback. This paper mainly focuses on the theoretical analysis and modeling of spherical arrays, following similar research methodologies, processes, and simulation conditions as References [29], [30], and [19]. Firstly, theoretical modeling based on continuous aperture is conducted, which is then discretized into sensor arrays. In Figure 14, we present the analysis results of the effects of discretization from continuous aperture to discrete arrays, showing minimal impact on the radiation pattern. The main contribution of this paper lies in the theoretical modeling and performance analysis applied to the early stage of array design. Various error factors in practical processes are not the main focus of this paper. There are dedicated studies on the coupling effects, such as mutual coupling, and channel amplitude and phase errors, affecting Dolph-Chebyshev and sum and difference patterns, including Bayliss difference patterns.
Point 3: The motivation behind the research is not explicitly communicated to a general audience. It is recommended to include additional diagrams to elucidate the problem and application scenarios, making the work more accessible and engaging.
Response 3: Thank you for reviewing our manuscript and providing valuable feedback. To address your concerns, we have added Figure 1 in the Abstract part to introduce the research motivation (Please see the Figure 1 and the paragraph above it, “This paper focuses on the mono-pulse technique applied to spherical sensor arrays. The mono-pulse technique is used for direction of arrival (DOA) estimation of a target. In Figure 1, the sum and difference patterns are symmetrical and anti-symmetrical about z axis. The ratio of the received signal to the sum and difference patterns varies with the angle between the target and the z-axis, which can be used for DOA estimation and tracking. The accuracy of DOA estimation is related to the width of the main lobe, and the narrower the main lobe, the higher the estimation accuracy. In order to reduce the impact of interference signals, both sum and difference patterns also need to meet low sidelobe constraints. The main lobe width mainly depends on the array size, the optimization range of this metric does not vary much for a defined array. Therefore, low sidelobes are often the goal pursued by various methods for sum and difference pattern synthesis, it is also the main focus of this paper.”
The above corrections we made are high lighted in the revised manuscript.
We hope that these revisions have enhanced the overall quality of our submission and addressed the concerns you raised. We look forward to receiving your further feedback and appreciate your invaluable guidance in helping us refine our work. Thank you once again for your constructive comments and invaluable guidance.
We sincerely hope you will be satisfied with the revised paper.
Best regards,
The Authors
Please see the attachment.

Reviewer 3 Report
Comments and Suggestions for Authors
Some details should be provided for enhancing the manuscript.
1) why is the pattern analysis?
2) optical images of the sensor array? What are the performances of the sensor array?
3) What kinds of the sensor array?
4) Comparsion with other sensors? or Other methods?
Author Response
Dear Reviewer,
We would like to express our gratitude for your thorough review of our paper. Your valuable insights have greatly contributed to the improvement of our manuscript. We have carefully revised the manuscript (ID: sensors-2887480, Title: On Difference Pattern Synthesis of the Spherical Sensor Array) based on your suggestions. We have diligently examined your perceptive comments and made the appropriate corrections, which we hope will meet your approval. The primary revisions in the paper include the following:
Point 1: Why is the pattern analysis?
Response 1: Thank you for reviewing our manuscript and providing valuable feedback. Is your problem related to pattern synthesis? We believe pattern synthesis is a critical step in array signal processing technology. It entails optimizing and adjusting the phase and amplitude of various sensor elements within the array to achieve the desired pattern. For instance, enhancing spatial anti-interference capabilities by generating patterns with low sidelobes. In the Abstract section, we have included Figure 1 to illustrate this topic. (Please refer to Figure 1 and the corresponding paragraph above it, “This paper focuses on the mono-pulse technique applied to spherical sensor arrays. The mono-pulse technique is used for direction of arrival (DOA) estimation of a target. In Figure 1, the sum and difference patterns are symmetrical and anti-symmetrical about z axis. The ratio of the received signal to the sum and difference patterns varies with the angle between the target and the z-axis, which can be used for DOA estimation and tracking. The accuracy of DOA estimation is related to the width of the main lobe, and the narrower the main lobe, the higher the estimation accuracy.…… ”)
Point 2: Optical images of the sensor array? What are the performances of the sensor array?
Response 2: Thank you for reviewing our manuscript and providing valuable feedback. We consider the main lobe width and sidelobe height as crucial performance metrics for sensor arrays. In the Abstract section, we have provided additional explanations regarding these metrics. (Please refer to Figure 1 and the accompanying paragraph above it for more details, “…… The accuracy of DOA estimation is related to the width of the main lobe, and the narrower the main lobe, the higher the estimation accuracy. In order to reduce the impact of interference signals, both sum and difference patterns also need to meet low sidelobe constraints. The main lobe width mainly depends on the array size, the optimization range of this metric does not vary much for a defined array. Therefore, low sidelobes are often the goal pursued by various methods for sum and difference pattern synthesis, it is also the main focus of this paper.”)
Point 3: What kinds of the sensor array?
Response 3: Thank you for providing valuable feedback. The proposed method is applicable to scalar sensor arrays, including microphone arrays or sonar arrays. Additional descriptions regarding this applicability have been provided in the Remark section below Equation (4). (Please refer to the paragraph below Equation (4) for more details, “Equation and are valid for scalar sensors (such as microphone sensors or sonar sensors) whose patterns exhibit rotational symmetry along the radial axis of the sensor. When accounting for mutual coupling effect, rotational symmetry properties can be approximately satisfied if the elements are distributed on the sphere’s surface according to the spherical t-design or the Coulomb design, refer to[32]. In this paper, the spherical t-design is specifically adopted. The rotational symmetry properties may not be satisfied in the practical application, such as ref. [2] and our previous work (ref. [26]), non-ideal factors in engineering can be obtained using electromagnetic simulation methods and the desired pattern and be synthesized using numerical iterative optimization algorithms, the methodology proposed in the paper can be applied to pre-theoretical designs and performance evaluation in the case.”)
Point 4: Comparison with other sensors? or other methods?
Response 4: Thank you very much for your valuable feedback on our manuscript.
In Fig.3 the Zolotarev difference pattern synthesis method for ULAs with odd numbered elements and spherical array proposed in the paper has been compared with the CVX method in ref. [35]. We have made modifications to the paragraph above Figure 3. “The figure confirms that both the proposed method and the numerical method in ref. [35] achieve the same optimum difference pattern with equal sidelobe.”
In Fig.4, the proposed Zolotarev difference pattern for ULA with odd numbered elements has been compared with ULA with even numbered elements. We have made modifications to the paragraph above Figure 4, “In Figure 4, the algorithm proposed in this paper synthesizes the Zolotarev pattern for ULA with 17 elements, while the method described in reference [16] synthesizes the Zolotarev pattern for ULA with 18 elements. The sidelobe constraint for both cases are set to 25 dB, and the polynomial utilized is the Zolotarev polynomial .”
In Fig.5, the proposed generalized Bayliss difference pattern for ULA with odd numbered elements has been compared with ULA with even numbered elements. We have made modifications to the text above Figure 5. “In the next example, the algorithm proposed in this paper synthesizes the generalized Bayliss pattern for ULA with 17 elements, while the method described in reference [16] synthesizes the generalized Bayliss pattern for ULA with 18 elements. The sidelobe constraint for both cases are set to 25dB.”
The above corrections we made are high lighted in the revised manuscript.
We hope that these revisions have enhanced the overall quality of our submission and addressed the concerns you raised. We look forward to receiving your further feedback and appreciate your invaluable guidance in helping us refine our work. Thank you once again for your constructive comments and invaluable guidance.
We sincerely hope you will be satisfied with the revised paper.
Best regards,
The Authors
Please see the attachment

Round 2
Reviewer 1 Report
Comments and Suggestions for Authors
The authors have addressed my concerns.
Author Response
Dear Reviewer,
We would like to express our gratitude for your thorough review of our paper. Your valuable insights have greatly contributed to the improvement of our manuscript.
Reviewer 2 Report
Comments and Suggestions for Authors
The revised version has addressed many of the questions. However, the manuscript can be improved by adding more discussions on the limitations of the proposed method and potential approach to address them.
Author Response
Dear Reviewer,
We would like to express our gratitude for your thorough review of our paper. Your valuable insights have greatly contributed to the improvement of our manuscript. We have carefully revised the manuscript (ID: sensors-2887480, Title: On Difference Pattern Synthesis of the Spherical Sensor Array) based on your suggestions. We have diligently examined your perceptive comments and made the appropriate corrections, which we hope will meet your approval. The primary revisions in the paper include the following:
Point 1: The revised version has addressed many of the questions. However, the manuscript can be improved by adding more discussions on the limitations of the proposed method and potential approach to address them.
Response 1: Thank you for reviewing our manuscript and providing valuable feedback. We appreciate the time and effort you have put into evaluating our work. To address your concerns, we have added discussions in the Conclusion part (Please see the last paragraph, “…The main contribution of this paper lies in the theoretical modeling and performance analysis applied to the early stage of array design, and the synthesis results can be used as the initial value of the numerical synthesis methods when dealing with various non-ideal factors. We focus on the common case of scalar sensor arrays such as microphone sensors or sonar sensors. When the vector spherical harmonics are adopted, the results can be extended to vector sensor arrays such as electromagnetic sensors.”
The above corrections we made are high lighted in the revised manuscript.
We hope that these revisions have enhanced the overall quality of our submission and addressed the concerns you raised. We look forward to receiving your further feedback and appreciate your invaluable guidance in helping us refine our work. Thank you once again for your constructive comments and invaluable guidance.
We sincerely hope you will be satisfied with the revised paper.
Best regards,
The Authors
